# Critical Factors of Promoting Design for Safety in China’s Subway Engineering Industry

**DOI:** 10.3390/ijerph17103373

**Published:** 2020-05-12

**Authors:** Yibo Yue, Xiaer Xiahou, Qiming Li

**Affiliations:** School of Civil Engineering, Southeast University, Nanjing 211189, China; yueyibo@seu.edu.cn (Y.Y.); xhcmre@seu.edu.cn (X.X.)

**Keywords:** design for safety, success factors, subway engineering, China

## Abstract

DfS (design for safety) can significantly reduce the engineering safety risk, decreasing the economic and social losses caused by safety accidents. To introduce DfS in China’s subway engineering, and with the help of a literature review and semi-structured expert interview, this study sorted out 28 promoting factors (including policy environment, practice, guidelines and tools, and education four categories) and seven impediment factors to DfS implementation. Through 80 questionnaire surveys of staff from universities, research institutes, and subway development, design, construction, and operation companies, this study screened out 26 promoting factors and seven impediment factors to assess the relationship between the two types of factors. The research provides a reference for DfS implementation in countries that have not implemented DfS before, such as China.

## 1. Introduction

With the scale and pace of China’s urbanization continues at an unprecedented rate, construction is becoming a more critical investment in the fixed national assets in urban service facilities by industry. For example, according to the statistics of the National Urban Rail Transit System of the Ministry of Housing and Urban-Rural Development of China (MOHURD), by the end of 2016, there were $272.63 billion of fixed national assets in urban service facilities. Among these urban service facilities, approximately 20% ($57.88 billion) was from the rail transit system (Table 1) [1].

However, accidents commonly occur during the construction and operation phases of a construction project. Taking the subway construction as an example, according to the MOHURD, approximately 246 accidents occurred during the 2002–2017 period [2], which resulted in a large number of casualties and economic losses. Hence, it is of the utmost importance to prevent or decrease safety accidents during the construction and operation stages. Currently, safety prevention and control cogitation are changing from passive acceptance to active prevention. Moreover, with the advancement of science and technology, the active prevention work is changing from “local prevention” to “comprehensive prevention,” from “pre-event prevention” to “advance prevention.”

In 1985, the International Labour Organization Pointed out that 60% of safety accidents occurred in engineering were related to design [3]. Therefore, research on Design for Safety (DfS) became a new direction for safety management [4]. DfS is also known as safety through design, prevention through design (PtD), design for construction safety (DfCS), and construction hazard prevention through design (CHPtD) [3,4,5,6,7]. Researchers believed that, although DfS is not the only factor affecting safety, it is a viable method of improving safety in construction sites [8,9], and taking precautions. With the advancement of research, some countries and organizations began to promote DfS. The 89/391/EEC directive issued by the European Union became the first DfS-related policy document. Subsequently, based on the 89/391/EEC directive and combined with their actual conditions, the European members promulgated and enacted relevant policies and regulations, such as Construction (Design and Management) Regulations 1994 (CDM 1994) formulated by the United Kingdom in 1994 [10]. Currently, the U.S., Europe, Singapore, Australia, and South Africa have successfully implemented DfS, and accident incident rates have already seen a definite decrease, but China has not yet. However, with the promotion of Building Information Modeling (BIM) and construction industrialization, DfS, as a representative of “advanced prevention,” will also be introduced, accepted, and promoted gradually in China.

According to the Engineering News Record (ENR), China’s construction enterprises have been taken the lead compared with other international rivals. In 2018, China’s construction enterprises made about $118.97 billion turnovers in the international market, which accounts for about 24.4% of the turnover from the world’s top 250 global contractors [11]. With the effect of China’s national reform strategies, such as the 13th five-year plan and the “Belt and Road,” there will continue to have significant growth on the overseas investment for China’s construction industry (CCI). By 2030, 65 countries between China and Europe will benefit in infrastructure construct, and the investment will be about $6 trillion [12]. Therefore, introducing DfS in China (especially in CCI) and promoting DfS worldwide through the overseas projects of China’s construction companies, are an essential step in the implementation of DfS, and the most important thing is how to introduce DfS based on experience from those succeed countries rapidly.

Researchers stated that understanding the impacts, obstacles, and facilitators of DfS implementation is helpful for institutions or organizations in accepting and spreading DfS [13]. Thereby, this research taking subway construction in China as an example analyzed the impediment and facilitating factors in CCI. The relationship between them, to facilitate CCI concentrate its efforts on the central issues in the process of promoting DfS and avoid detours. Moreover, the research can provide a reference for countries or researchers interested in DfS.

## 2. Literature Review

Presently, the research related to DfS is changing from the basic research of previous years to more recent, applied research. Previously, research on DfS mainly focused on the concept of DfS [5,7,14] and legal liability brought by DfS [15], the relationships between DfS and accident rates [16,17,18], the relationships between the implementation of DfS and stakeholders respectively, such as owners [13,14,19], designers [19,20], contractors [16,21], the relationship with the contract [22], the relationship with social sustainability [23,24], and the relationship with guidelines and tools [25,26,27]. However, nowadays, most international researches related to DfS are aimed at combing DfS with BIM or other information technologies [28,29]. Moreover, some of the researchers focused on implementing DfS in the risk management of a specific category of engineering. Borchiellini, et al. [30] confirmed the effectiveness of the DfS approach to improving the safety and health conditions in underground mining activities. Based on two cases, Labagnara, et al. [31] confirmed that introducing the DfS approach in tunnel driving can not only improve the flexibility of construction but also avoid unscheduled stops and reduce costs caused by claims.

Currently, there are also many types of research related to impediment factors or facilitating factors of DfS. In terms of the impediment factors, some researchers believed that the lack of laws and regulations requiring that designers consider the safety of construction workers hinder the spread of DfS [13,16,32,33]. Some researchers focus on the unclearness of the designers’ safety responsibility [34,35,36]. Toole [37] found that the lack of communication between designers and other stakeholders was one of the obstacles for DfS, which was verified by other researchers [32,38,39]. Further, researches revealed that the lack of DfS knowledge and related training for designers [40,41,42,43,44], the lack of DfS guidelines and tools [35,45,46], increased the design cost [39,47,48]. All these factors will affect the implementation of DfS.

For the facilitating factors, some researchers proposed from the aspect of education [6,9,20,22,49,50,51]. Popov, et al. [52] believes that it is beneficial for DfS to encourage university and research institutions to conduct DfS related research and to introduce certified professional security personnel into the college classroom. Some researchers believed that promulgating and enacting DfS related laws and regulations, helped create the DfS culture atmosphere from the whole industrial chain [10,40,45,53]. Some researchers advocated developing DfS toolkits and guidelines [13,29,32,43,54,55] or increasing communication between designers and other stakeholders [16,56,57] to facilitate the implementation of DfS.

Overall, for the one part, the existing research mainly focuses on only a single perspective, like owners, designers, contract, safety liability. For the other, present researches related to DfS factors mainly based on countries that have implemented DfS. Instead of countries that have never implemented DfS, like China.

## 3. Research Objective and Methodology

### 3.1. Research Objective

The propose of this research is to identify the impediments and facilitating factors, and the relationships between the two factors, to help CCI introducing DfS successfully. The research combined the qualitative method with quantitative research. The qualitative methods, including a literature review and expert interviews, which provide a means for discovering and understanding the meaning of particular human behavior or social problem caused by an individual or group [58,59]. The quantitative method in this study is a questionnaire survey, which is useful in obtaining a large amount of data in a short period and widely used to evaluate critical practical factors or prevalence in an industry. The research divided into two parts. The first part is the identification of impediments and facilitating factors, which mainly uses the method of literature review, expert interview, and questionnaire survey. The other part is the identification of the relationship between the two factors, mainly using the questionnaire survey method.

### 3.2. Factor System Screening

#### 3.2.1. Preliminary Factor Screening

Two methods used for the acquisition of preliminary factors. The first one is intensively analysis of literature from Web of Science and www.cnki.net, with keywords like the design for safety, design by safety, safety in design, design for construction safety, prevention through design, and safety through design. Also, the study supplemented literature analysis according to Prevention through Design in the Construction Industry, published by Bucknell University in 2017. Due to the limitation of data resources download in Southeast University Library, 242 related kinds of literature downloaded in total, including 20 kinds of literature in Chinese. The other one is a summary of knowledge and experience from research team members. Based on in-depth literature analysis, combined with the knowledge and experience of team members, the study obtained a preliminary list of impediments and facilitating factors for the implementation of DfS in China.

The expert interview method can quickly and objectively integrate the experience and subjective judgment for factors of most experts, which provide a basis for the followed questionnaire design. In terms of expert selection, some research listed the essential elements [60,61]. Combing relevant perspectives, the research selected expert according to the following principles: (1) have more than ten years of practical or academic experience in the fields of engineering management, subway engineering design, and safety risk management; (2) have a master’s degree or above; (3) be a member of the Architectural Society of China; (4) have time and willingness to participate in this interview; (5) highly objective and reasonable. Finally, selected five experts for this study; coming from subway design company (1 person), subway construction company (1 person), subway operating company (1 person), software developing company (2 people), and university and research institutes (2 people) respectively.

The expert interview in this research conducts a semi-structured interview. That is, after selecting the experts, firstly, the research listed the factors which sorted out by the literature review and research members’ experience so that the experts can easily choose factors what they think is the obstacles and facilitator. Secondly, consulting experts’ overall views on these factors and whether there are factors that need to be adjusted. If the expert disagreed with a specific factor as an obstacle or facilitator, it should explain the reason. Finally, counting the results of experts’ selection.

For this study, the researchers used the following criteria to retain or omit elements: if the expert retained the factor, scored “1”; otherwise, scored “0”. Finally, the research will retain elements with a level of agreement of 60% (more than three experts), and omit elements were under the level of 60% agreement. The retained elements considered to be very reasonable. After an expert interview, the research acquired the preliminary impediment factor system and facilitating factor system.

#### 3.2.2. Questionnaire Survey and Final factor screening

##### Content of Questionnaire Survey

At the beginning of the questionnaire, there is a description and instructions. Including the purpose of the survey, the survey team, and the concept of DfS. The content of the questionnaire divided into five parts.

The first section is a primary information survey, including six multiple-choice questions, which regarding the respondent’s age, the educational background, the type of work conducted within the respondent’s firm, the personal work experience, and two questions about their attitude and awareness on DfS. All the questions in this section are multiple-choice questions.

The second section is about whether a certain factor can obstruct the implementation of DfS in CCI, so we designed it as Likert scale questions. As 5-, 7-, and 10-scale questions will get the same mean in the same survey [62], 5 points scales were chosen in the research, in which 1 = extremely disapproval, 2 = more disapproval, 3 = general approval, 4 = more approval, and 5 = extremely approval.

The third section is about whether a particular factor can take a facilitating effect on the implementation of DfS in CCI, and designed as 31 Likert scale questions.

The fourth section is about whether one facilitating factor can affect or change the statement of impediment factors, respectively. Questions in this section are multiple-choice questions. The investigator can choose more than one impediment factor in the question.

The fifth section is an open-ended question, mainly related to respondents’ attitudes of DfS and the questionnaire.

##### The Selection of Investigators

The sample investigators selected in three ways: (1) For the investigators from academic, the researchers browsed the official websites of the top 30 universities and research institutes in China, selected and mailed investigators from the field of subway or rail transportation engineering industry. In total, 140 questionnaires sent. (2) With the help of WeChat and QQ, distributing questionnaire links to relevant professional groups field in rail transit (each group with members between 300–500), inviting people in the group to fill out and to spread the questionnaire. (3) Through the relationship of the research team, inviting individuals in the well-known rail transit companies and infrastructure construction companies filling out and spreading the questionnaire.

The survey conducted from May 1st to June 30th, 2017. And through the online questionnaire platform. Also, to ensure the quality of the survey, the questionnaire was designed that the same investigators can only fill out the questionnaire once through a computer or mobile phone.

##### The Final Factor System Screening

According to the statistical analysis results of the questionnaire survey, the research omitted factors with a mean less than 3, and form the final impediment factor system and facilitating factor system.

#### 3.2.3. Identification of Relationships between Impediment Factor and Facilitating Factor

According to the statistical results of the questionnaire survey, if more than 60% of investigators approval the relationship between two factors, they were deemed to be related. Otherwise, they were unrelated.

## 4. Research Results

### 4.1. Preliminary Screening of Indicators

After in-depth reading and analyzing 242 English kinds of literature, 28 articles involved content related to impediment factors and 63 articles related to facilitating factors. While the analysis of 16 Chinese pieces of the literature revealed that only three articles related to impediment, and two related to facilitating factors. Identified twenty-three impediment factors and 48 facilitating factors. Based on the knowledge reserve of research members, the preliminary factors amended as follows. (1) Combining similar factors or factors with the subordinated relationship, such as combined the factor lack of DfS knowledge for designers with the factor lack of construction experience for designers. (2) Removed factors that are inconsistent with the status quo of Chinese engineerings, such as the US construction industry, health, and human service offices, and homeowners, consortia, and safety experts, recognize the importance of DfS.

In summary, 31 facilitating factors and seven impediment factors were retained (see Table 1 and Table 2 for detail). Also, according to the connotation of the facilitating factors, the researchers classified them, namely the policy environment, the practice, the guidelines and tools, and the education. The following table lists the coding of specific factors, their connotations, the related literature, and the frequency in the literature.

### 4.2. Expert Interview

After the expert interview, seven impediment factors and 28 facilitating factors retained. The detailed results of the expert interviews seeing in Table 3, in which black means the factor retained, and red means the factor omitted. Among the impediment factors, although seven factors retained, three experts held that the connotation of (3) in I2 and (3) in I6 did not conform to the actual situation in China, for this reason, 2 of the experts chosen omitting the factors. In I7, one of the experts who come from the research institutes believes that although the factor can affect the step of China’s implementation of DfS, it cannot be the main factor. While among the facilitating factors, F29 and F39 omitted for three experts disagree with their influence, and F48 omitted because four experts deemed that the factor was not suitable for countries without a DfS project like China.

### 4.3. Questionnaire Survey

To ensure the feasibility of the questionnaire, the researchers randomly selected five practitioners from the relevant industry practitioners to conduct a pre-investigation. The results of the pretest showed that the questionnaire was feasible.

Eighty-four survey responses were collected. However, four questionnaires discarded (effective questionnaire rate is 95.24%), three of them were incomplete, and one of the responders was not work in the related professions. For the questionnaire was disseminated both via the internet and the email, the size of some of the targeted groups that received the request for participation was unknown, the researchers were not able to determine the exact response rate for the survey cannot accurately calculate.

From the perspective of basic information, 26 (32.5%) of respondents from design units (see Table 4 for detail). In terms of educational background, 78 (97.5%) of them at least have a bachelor′s degree (see Table 5 for detail). Also, all the owners and researchers among the responders hold a master′s degree or above.

About the age of the responders, 43 (53.75%) of them are older than 30, and 35(43.75%) of them are 26–30 years old. When it comes to working years, 21 (26.25%) responders have more than 11 years of working experience, and only 25 (31.25%) responders have less than five years of working experience (see Table 5 for detail). Moreover, only two responders versed in DfS and have relevant research or work experience, and it is the first time for 76.25% (61) of responders to accept knowledge about DfS. However, more than 78.75% of them believe that it is feasible to implementing DfS in the CCI (see Table 6 for detail). Further, 17 respondents who disapproved of the effectiveness of DfS in CCI are mainly researchers, designers, and general contractors.

#### 4.3.1. Reliability Analysis

Reliability analysis, which refers to the consistency and stability of measurement results, can be detected by the reliability coefficient. The variation range of the reliability coefficient is between 0–1, the higher the coefficient value, the higher the stability and reliability of the detection, and the more acceptable of the result. Cronbach α is undoubtedly one of the most essential and pervasive statistics indicators in research involving test construction and use. At the same time, Cortina [63] states that the threshold value of Cronbach α for a high reliable questionnaire is 0.8. The reliability analysis of 80 valid questionnaires in this study was carried out by SPSS 22.0. Moreover, the Cronbach α coefficient values were 0.912. For the aspect of the amended terms’ total correlation (CITC) and the deleted terms’ α coefficient, all the terms’ CITC is above 0.3. All deleted terms’ α coefficients are not significantly higher than the α coefficient, so there is no delete term during analysis, which means the scale has excellent reliability and strong stability.

Validity analysis is the extent to which the measurement of the scale can represent or reflect the targeted construct [64], that is, whether the test result of the questionnaire can reflect the degree of the objective reality that should reflect. Currently, Kaiser–Meyer–Olkin (KMO) and Bartlett sphere used to judge the validity of the questionnaire. If the KMO value is higher than 0.8, the validity is suitable. If the significance value (sig. value) is less than 0.500, the null hypothesis should reject, which means the structure validity of the questionnaire is acceptable. The study conducted a validity analysis using SPSS 22.0. The KMO test results were 0.883, more significant than 0.8. At the same time, the Bartlett sphere test was significant at *p* = 0. The two index standards indicated that the questionnaire of this study met the criteria of validity.

#### 4.3.2. The Final Index System

The statistics results of the questionnaire showed that the responders approbated almost all the facilitating factors and impediment factors set in the questionnaire. In this step, using the following criteria to omitted factors: when the overall mean of a factor is less than 3, and the coefficient of variation is more than 15%, it is considered to be unsuitable. It should be omitted, like the F18 and F28. In the end, there are seven impediment factors and four categories (26 factors) facilitating factors retained in the research (see Table 7 and Table 8 for detail, in which black means the factor retained, and red means the factor was omitted).

#### 4.3.3. The Relationship between the Facilitating Factors and Impediments

For the relationship between the facilitating factors and impediments, the research assigns 0 and 1 to responder’ choices, that is, if the responders believe that there is a relationship between the two factors, it was assigned 1, otherwise, assigned 0. In the end, if the total assigned Number was more than 48 (60% of responder approval the relationship), they are considered to be related. Otherwise, they are not related. According to the criteria, the relationship between impediment factors and facilitating factors is shown in Table 9 (in which black means the two factors related, otherwise means unrelated) and Figure 1.

## 5. Research Analysis

Combing with the basic information of responders, the research summarizing and analyzing the results from 5 aspects (policy environment, practice, guidelines and tools, education, and impediments). Figure 2 showed the influence degree of each facilitating factor on impediments, respectively, in which 0, 20, 40, 60, and 80 each means 0, 20, 40, 60, and 80 respondents believe the two factors have facilitating relationship. Moreover, the reference line means the facilitation boundary, outside of the reference line means the facilitating factor can improve the impediment. Otherwise, the facilitating factor has little facilitating influence on the impediment.

### 5.1. Policy Environment

Compared with other countries, China has yet to implement DfS. Therefore, even though the questionnaires sent to the person who works in a subway engineering or safety-related industry, more than 97% of respondents have little knowledge about DfS, and even more than half of them had never heard of DfS. The only two respondents who were familiar with DfS completed the questionnaire through email (which means the respondents were researchers in research institutes or universities engaged in the related research), and their IP was in America. Therefore, it is crucial and necessary for the government and industry associations to introduce and publish some related policies and measures firstly. Like in Singapore, the government firstly formulated the Work Safety and Health (Safety Design) Regulations (2015 Edition) (WSH (DFS) 2015) and subsequently promoted in practical projects [65]. According to the results of the questionnaire survey, the policy environment categories’ overall mean score rank in the whole facilitating factor categories also verified this.

All the sub-contractors and consultants in the survey believe that the implementation of DfS is helpful to the safety management of CCI, and it is worth promoting. Respondents from construction Co. (13 respondents totally) have two extreme perceptions on DfS, five people thought it was necessary to implement DfS, but five people disagree. Most of the respondents from the university and research institute and design Co. choose to depend on the situation. However, researchers, designers, and general contractors are the most critical three kinds of person for the development of the DfS tool, and F14 and F18 are two factors which for the propose of changing the perceptions and attitudes of the society, thereby increasing their initiative to promote DfS. Their different view on DfS directly leading the low average mean of F14 and F18. Furthermore, F18 ranked 28 in the mean, is omitted for the mean lower than 3.0. Also, it cannot be ignored that F14 is the most influential factor in impediment, together with F13, and it can improve about six impediment factors.

Also, as facilitating factors in policy environment are relevant to not only designers but also the whole stakeholders, five (71.43%) of the seven factors rank in the top 10. At the same time, all the facilitating factors in this category can affect more than half of impediments, and F13 can be capable of all the impediment factors.

### 5.2. Practice

Previous scholars have suggested that if relevant stakeholders (especially owners and designers) understand DfS, they will be willing to change the status quo [4,66], which verified in this study. As a result of 21.25% responders (mainly came from design Co., construction Co., and research institute) holding that, there is no need to implement DfS in China. The rank of facilitating factors in the practice category in the overall mean score was affected, and the group means ranked for four. At the same time, as DBB is the most commonly used model in China’s construction, a construction expert can only join the project after all the design completed. Caused the degree of recognition of F28 by respondents varies greatly, and leads to F28 omitted for the mean lower than 3.0. Nevertheless, some respondents in this survey believed that practice is an indispensable step for the implementation of DfS in China.

In terms of the relationship with the impediments, the F2 factors mainly focus on designers and design companies, factors F2 mainly make a difference in I3 and I6, namely, six among 7 F2 factors can improve the status of I3 and I6. Moreover, F26 would have an impact on all of the impediments except I2. At the same time, F21, F23, and F24 can all improve half of the impediments.

### 5.3. Guidelines and Tools

Nowadays, the review of design safety in China mainly focuses on whether the design drawings comply with the laws, regulations, and standards, that is, the compliance review for the design drawings. Even though the use of BIM had some help, mostly the review dependent on the subjective judgment of safety experts. Compared with the existing tools in other countries, like the DFCS toolbox [10,65], the CHAIR [27], the DFSP tool [67], there is still a large gap, therefore, the research promoting five factors in this category.

Compared with other categories, there are relatively few facilitating factors related to guidelines and tools, only five factors, and the F35 is learning techniques and methods from other industries. However, guidelines and tools considered to be the most indispensable part of the DfS research, 22 of the literature (which is the most of all the promoting factors in the amount of literature) in this study, suggested that developing appropriate DfS tools is conducive to the implementation of DfS. Besides, although the category ranked third in the overall mean score, the mean score rank for each factor in this category is relatively low, except F35, the other factors in this category all ranked behind 14 (26 factors in total). As these factors are more targeted, when it comes to the impact on impediments, almost all the factors can improve more than four obstacles, except F35, which can only make an impact on I3.

### 5.4. Education

Currently, countries that have successfully implemented DfS have implemented many practices in education. For instance, most of the civil engineering courses in the UK included DfS modules [68], Australia compiled the DfS education resource packs [69], Singapore established the WSHO certification system [65], and the DFSC competition set by the USA [70]. Also, the related countries have launched 1–3 days of training programs for on-the-job designers [70]. However, for DfS has not formally implemented DfS, there are little DfS education resources in China, and not only the students and designers but also the faculty lack of DfS knowledge. To this end, the research set seven educational facilitating factors.

The facilitating factors in the category of education are relatively specified; the group means reached 4.029, ranked second. However, for the education-related factors that are more targeted in the improvement of impediments, factors in this category can only affect 2–3 impediments, except F47, which can change the situation of I2, I3, I5, I6, and I7.

Moreover, when conducting the expert interviews, two of the experts thought that, although related researchers believe that taking DfS education for all the stakeholders, especially designers and owners, is helpful for the implementation of DfS, there has no unified understanding in education for students. Some researchers believed it is better to provide related courses separately, while some scholars believe that related education can form a separate module throughout the design or construction courses. As there has been no individual course study for DfS in China, implementing the DfS course in China can start with adding some DfS modules to relevant compulsory courses.

### 5.5. Impediments

Although facilitating factors in policy environment group is most likely to promote the step of DfS in CCI, the impediment factor I1, namely lacking regulations and norms for DfS, only is ranked 4 in the mean score rank. Also, among all the literature, the factor I3 has 20 literature sources, ranking first, the same with the rank of factor mean, which means the result was the same with other researchers’ recognition about DfS knowledge. However, for factor I5, ranking fifth in literature sources, however, ranked second in the mean. Experts stated that, among the impediment factors, the first three factors in the mean rank, means the respondents have a high recognition, and the three factors can be easily identified, compared with other factors. Further, I1, I3, and I5 are the first three impediment factors for most designers. However, for respondents from university and research institutes, the first six factors are almost the same important.

For all the impediment factors, I3 had the most improving methods, while I2 had the least improved proposals. Specifically, all the facilitating factors in the policy environment group can make the change on I1 and I6, while all the facilitating factors in F3 and F4 can affect the situation of I3. Also, all the factors in the education group did not affect I1 and I4.

## 6. Conclusions

This study uses qualitative and quantitative methods to study the key obstacles and facilitating factors for the promotion of DfS in CCI, and the relationship between the two factors. The research revealed that designers’ lack of DfS related knowledge, together with a lack of guidelines and tools, are the biggest obstacles for DfS implementation. However, among the facilitating factors, those related to the policy environment are more likely to open the process of DfS promoting, and influence the obstacles. Especially for F13, which can influence all the impediment factors. The accumulation of DfS knowledge is a multifaceted process. Therefore, almost all the facilitating factors can more or less change the insufficient DfS knowledge reserve.

Although The research focuses on the CCI, most of the relevant data sources are from subway related companies. In the following research, the sample capacity should be expanded, increasing investigated personnel from different types of projects, and focusing on the difference among different project types. Most of the impediments and facilitating factors in this research considered from the perspective of designers. However, the responses are from all stakeholders. In the future, research can focus on analyzing from the perspective of a single stakeholder.

## Figures and Tables

**Figure 1 ijerph-17-03373-f001:**
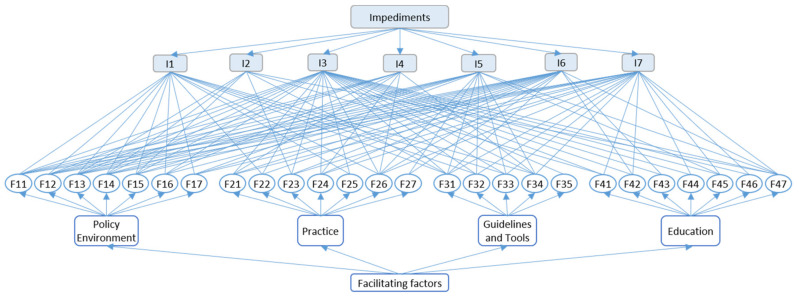
The relationship between impediments and facilitating factors.

**Figure 2 ijerph-17-03373-f002:**
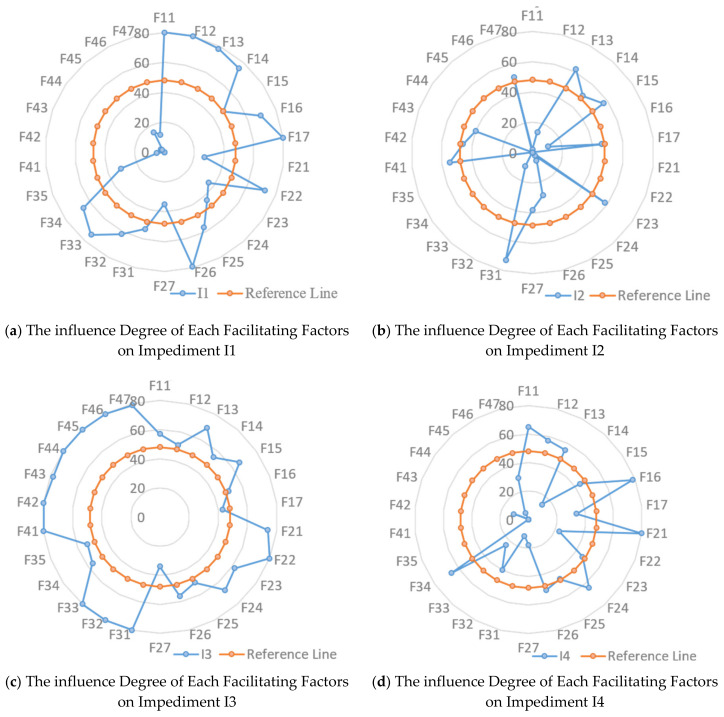
The Influence Degree of Each Facilitating Factors on Impediments, Respectively.

**Table 1 ijerph-17-03373-t001:** The preliminary impediment factors.

Code	Impediments Factors	Connotation of Impediment Factors	Total No. of Reference
I1	Lackness of regulations and norms for DfS	(1) Although some of current regulations and norms require designers to consider safety, the effect is not apparent; (2) there is no specific DfS regulations or mandatory DfS clause	6
I2	Unclearness of safety responsibilities for designer	(1) The current laws and regulations are unclearly in designers′ safety responsibility on construction and operation phase; (2) the responsibilities of designers′ in safety accidents is difficult to quantify and value; (3) the lawyer′s advice designers do not consider of design safety, to avoid or decrease sue	11
I3	Insufficiency of designers′ knowledge reserves	(1) Most of the designers work in the design company directly after being graduate, which means they lack experience from a construction site; (2) DfS knowledge model is insufficient in designers′ college education; (3) shortage of education resource and training resource specifically for designers′ DfS education	20
I4	Lackness of Communication between designers and other participants	(1) designers′ job is relatively independent, and most of the designers only count for one part in a project, leading lack of conscious of overall safety design for the whole project; (2) traditional project delivery model separate the whole life cycle of a project, leading of less communication between designers and other participants	15
I5	Lackness of DfS tools	(1) Other countries in the world developed most of the Tools, leading lack of localization DfS tools in China; (2) the existed tools are complicated in operation, intelligent, and even some of them can only use in the specify safety risk	7
I6	Insufficiency of motivation for designers to implement DfS	(1) developers did not require the implementation of DfS when the project was tending; (2) design cost will increase with the implementation of DfS; (3) the workload of designers in China tends to be saturated, the implementation of DfS will increase their workload	9
I7	Others	(1) It is challenging to conduct research related to health, leading less research related to DfS, and cannot take into practice; (2) there is not complete spread network for DfS knowledge	2

**Table 2 ijerph-17-03373-t002:** The preliminary facilitating factors.

Category	Code	Facilitating Factors	Total No. of Reference
Policy Environment (F1)	F11	Drafting and promulgating national-level DfS strategic plan	2
	F12	Drafting and promulgating relative regulations and policies documents, such as financial incentives, and tax credits or reduction measures to stimulate developers and design enterprises promote DfS	11
	F13	Encouraging the industry associations to compose or compile wide-ranging, in everyday use, and consensus industry standards	2
	F14	Encouraging universities and research institutes to conduct DfS related research, especially research related to the concept and the application of DfS	5
	F15	During the bidding, encouraging the owners explicitly request design companies conduct design contain DfS or safety design reviews	9
	F16	Promote delivery models like DB\EPC\IPD	6
	F17	Setting up pilot demonstration projects or provinces, advocating government buildings or government-invested buildings adopted design proposal contain DfS	8
	F18	Modified the contract template, actuate owners exempt or release designers′ legal liability caused by conducting DfS during the design	2
Practice (F2)	F21	Encouraging cooperation and communication between designers and other participants during the design	10
	F22	Setting up DfS certification for personals and organizations	3
	F23	Collecting information about DfS related project	6
	F24	Setting up awards for personal or organization, such as” Best Practice” and “Best Solution.”	2
	F25	Encouraging designers increase the usage of prefabricated products and decrease the usage amount of little hazard materials and systems in the project	3
	F26	Adding DfS related index into the Evaluation Criteria of Green Building	3
	F27	Request designers take the whole projects′ safety design plan into their consideration	2
	F28	Encouraging experienced contractor participant in the phase of the design	6
	F29	Compose safety-related design files	2
Guidelines and Tools (F3)	F31	Composing and compiling DfS related instruction manuals or design guidelines	7
	F32	Constructing a data-sharing platform similar to the knowledge base or data processor	4
	F33	Developing usable or referred safety tools for designers according to the related laws and regulations, and norms in China,	22
	F34	Developing usable or referred safety tools for designers according to the related laws and regulations, and criteria and norms in China	4
	F35	Learning technology and methods from other industries	1
	F36	Setting up a quantified index evaluation system for safety design	1
Education (F4)	F41	Formulating a national-level educational plan, including PTD research plan, designer training plan, education resources complying plan, and faculty training plan	8
	F42	Composing teachers′ handbooks, brochures, short-term training materials, and undergraduate textbooks. those textbooks should include the PTD model, precautions for workers′ safety and health, or especially hazards during the phase of construction design	3
	F43	Opening DfS short-term train course for faculty and designers	11
	F44	Adding DfS knowledge model and construction cases into the obligatory professional course of design-related discipline in colleges and universities	15
	F45	Launching various forms like a seminar, lecture, and subject competition to help design-related graduates understanding and applying of safety theory	6
	F46	Encouraging professional safety engineers and national certificated safety professionals offer DfS related opening courses or lectures with the help of network course platform like MOOC	1
	F47	Encouraging researchers and designers were diffusing and protecting DfS knowledge through journals, websites, and news. Organizing specialized and DfS-revolved seminar, forum, and academic conferences	2
	F48	Setting up of DfS social practice base	2

**Table 3 ijerph-17-03373-t003:** The result of the expert interview.

Dimension	Indicator	Score	Verification	Dimension	Indicator	Score	Verification
Impediments	I1	5	Retained	Facilitating Factors—Practice	F25	5	Retained
	I2	3	Retained		F26	5	Retained
	I3	5	Retained		F27	5	Retained
	I4	5	Retained		F28	5	Retained
	I5	5	Retained		F29	2	Omitted
	I6	3	Retained	Facilitating Factors—Guidelines and Tools	F31	5	Retained
	I7	4	Retained		F32	5	Retained
Facilitating Factors—Policy Environment	F11	5	Retained		F33	5	Retained
	F12	5	Retained		F34	5	Retained
	F13	5	Retained		F35	5	Retained
	F14	5	Retained		F36	2	Omitted
	F15	5	Retained	Facilitating Factors—Education	F41	5	Retained
	F16	5	Retained		F42	5	Retained
	F17	5	Retained		F43	5	Retained
	F18	5	Retained		F44	5	Retained
Facilitating Factors—Practice	F21	5	Retained		F45	5	Retained
	F22	5	Retained		F46	5	Retained
	F23	5	Retained		F47	5	Retained
	F24	5	Retained		F48	1	Omitted

Noted: black means the factor retained, and red means the factor was omitted.

**Table 4 ijerph-17-03373-t004:** Statistic of the nature of respondents’ work.

Nature of Work	Government Department	Design Unit	Project Co. (Investor)	General Contractor	Consulting Unit	Subcontractor	University/Research Insitute	Totally
Number	7	26	7	13	6	5	16	80
Percentage	8.75%	32.5%	8.75%	16.25%	7.5%	6.25%	20%	100%

**Table 5 ijerph-17-03373-t005:** Statistic of the age, educational background, and working years of the respondent.

Age	No.	Percentage	Education Background	No.	Percentage	Working Years	No.	Percentage
<20	0	0.00%	Below College	0	0.00%	<1	0	0.00%
21–25	2	2.50%	College	2	2.50%	2–5	25	31.25%
26–30	35	43.75%	Bachelor Degree	30	37.50%	6–10	34	42.50%
31–35	19	23.75%	Master Degree and above	48	60.00%	>11	21	26.25%
36–45	17	21.25%						
46–55	4	5.00%						
>56	3	3.75%						
Totally	80	100.00%		80	100.00%		80	100.00%

**Table 6 ijerph-17-03373-t006:** Statistic of Cognitive and attitude for DfS (Totally Number = 80).

Question	Choice	Number	Percentage
Do you know about DfS?	Do not know, and no idea	61	76.25%
	Have some understanding, but do not go deep	17	21.25%
	Very knowledgeable, and have relevant research or work experience	2	2.50%
Do you agree that DfS can implement in the Chinese construction industry?	Totally disapproval, not necessary or feasible	17	21.25%
	General approval, depending on project conditions	44	55.00%
	Extremely approval, DfS should be encouraged and promoted	19	23.75%

**Table 7 ijerph-17-03373-t007:** Scores and ranking of the impediment factors.

Impediment	Mean	Standard Deviation	Coefficient of Variation	Rank	Verification
I1	3.73	0.48	12.87%	4	Retained
I2	3.51	0.41	11.68%	5	Retained
I3	4.13	0.31	7.51%	1	Retained
I4	3.86	0.32	8.29%	3	Retained
I5	4.01	0.45	11.22%	2	Retained
I6	3.23	0.39	12.07%	6	Retained
I7	3.13	0.26	8.31%	7	Retained

**Table 8 ijerph-17-03373-t008:** Scores and ranking of the facilitating factors.

Dimension	Facilitating Factors	Mean	Standard Deviation	Coefficient of Variation	Rank of Mean	Group Mean	Group Rank	Verification
Policy Environment (F1)	F11	4.76	0.36	7.56%	1	4.1938	1	Retained
	F12	4.69	0.4	8.53%	3			Retained
	F13	4.35	0.31	7.13%	9			Retained
	F14	3.71	0.5	13.48%	19			Retained
	F15	4.39	0.46	10.48%	6			Retained
	F16	4.21	0.43	10.21%	11			Retained
	F17	4.54	0.45	9.91%	5			Retained
	F18	2.9	0.76	26.21%	27			Omitted
Practice (F2)	F21	3.94	0.38	9.64%	16	3.7588	4	Retained
	F22	3.86	0.48	12.44%	17			Retained
	F23	4.15	0.51	12.29%	13			Retained
	F24	4.37	0.44	10.07%	8			Retained
	F25	3.49	0.46	13.18%	22			Retained
	F26	4.19	0.45	10.74%	12			Retained
	F27	3.16	0.39	12.34%	26			Retained
	F28	2.91	0.71	24.40%	28			Omitted
Guidelines and Tools (F3)	F31	4.39	0.47	10.71%	6	3.826	3	Retained
	F32	3.57	0.37	10.36%	21			Retained
	F33	4.04	0.46	11.39%	14			Retained
	F34	3.7	0.38	10.27%	20			Retained
	F35	3.43	0.31	9.04%	24			Retained
Education (F4)	F41	4.75	0.26	5.47%	2	4.0286	2	Retained
	F42	3.96	0.56	14.14%	15			Retained
	F43	4.57	0.45	9.85%	4			Retained
	F44	4.26	0.47	11.03%	10			Retained
	F45	3.36	0.44	13.10%	25			Retained
	F46	3.83	0.43	11.23%	18			Retained
	F47	3.47	0.34	9.80%	23			Retained

Noted: black means the factor retained, and red means the factor was omitted.

**Table 9 ijerph-17-03373-t009:** Scores and the percentage of the relationship between impediments and facilitating factors.

Facilitating Factors	Impediments (No./Percentage)						
	I1	I2	I3	I4	I5	I6	I7
F11	80 (100%)	2 (2.5%)	57 (71.25%)	65 (81.25%)	51 (63.75%)	69 (86.25%)	77 (96.25%)
F12	80 (100%)	14 (17.5%)	51 (63.75%)	57 (71.25%)	54 (67.5%)	80 (100%)	49 (61.25%)
F13	78 (97.5%)	62 (77.5%)	69 (86.25%)	55 (68.75%)	78 (97.5%)	70 (87.5%)	51 (63.75%)
F14	75 (93.75%)	50 (62.5%)	55 (68.75%)	14 (17.5%)	62 (77.5%)	78 (97.5%)	80 (100%)
F15	48 (60%)	57 (71.25%)	66 (82.5%)	44 (55%)	63 (78.75%)	75 (93.75%)	46 (57.5%)
F16	69 (86.25%)	11 (13.75%)	50 (62.5%)	78 (97.5%)	23 (28.75%)	50 (62.5%)	34 (42.5%)
F17	80 (100%)	46 (57.5%)	43 (53.75%)	34 (42.5%)	67 (83.75%)	80 (100%)	50 (62.5%)
F21	27 (33.75%)	0 (0%)	74 (92.5%)	80 (100%)	46 (57.5%)	66 (82.5%)	62 (77.5%)
F22	72 (90%)	0 (0%)	80 (100%)	23 (28.75%)	40 (50%)	67 (83.75%)	23 (28.75%)
F23	36 (45%)	58 (72.5%)	62 (77.5%)	46 (57.5%)	34 (42.5%)	74 (92.5%)	74 (92.5%)
F24	43 (53.75%)	3 (3.75%)	67 (83.75%)	64 (80%)	34 (42.5%)	80 (100%)	69 (86.25%)
F25	57 (71.25%)	6 (7.5%)	51 (63.75%)	47 (58.75%)	46 (57.5%)	46 (57.5%)	42 (52.5%)
F26	79 (98.75%)	29 (36.25%)	56 (70%)	51 (63.75%)	55 (68.75%)	78 (97.5%)	65 (81.25%)
F27	35 (43.75%)	38 (47.5%)	34 (42.5%)	18 (22.5%)	57 (71.25%)	51 (63.75%)	24 (30%)
F31	53 (66.25%)	73 (91.25%)	80 (100%)	12 (15%)	74 (92.5%)	67 (83.75%)	74 (92.5%)
F32	62 (77.5%)	10 (12.5%)	80 (100%)	40 (50%)	80 (100%)	45 (56.25%)	80 (100%)
F33	74 (92.5%)	0 (0%)	80 (100%)	24 (30%)	80 (100%)	40 (50%)	80 (100%)
F34	66 (82.5%)	0 (0%)	56 (70%)	66 (82.5%)	65 (81.25%)	41 (51.25%)	79 (98.75%)
F35	31 (38.75%)	0 (0%)	53 (66.25%)	0 (0%)	29 (36.25%)	43 (53.75%)	34 (42.5%)
F41	5 (6.25%)	55 (68.75%)	80 (100%)	0 (0%)	14 (17.5%)	45 (56.25%)	80 (100%)
F42	0 (0%)	46 (57.5%)	80 (100%)	0 (0%)	33 (41.25%)	80(100%)	80 (100%)
F43	0 (0%)	40 (50%)	78 (97.5%)	11 (13.75%)	3 (3.75%)	57 (71.25%)	56 (70%)
F44	3 (3.75%)	0 (0%)	80 (100%)	0 (0%)	42 (52.5%)	34 (41.5%)	80 (100%)
F45	2 (2.5%)	0 (0%)	80 (100%)	0 (0%)	72 (90%)	40 (50%)	80 (100%)
F46	15 (18.75%)	0 (0%)	80 (100%)	5 (6.25%)	22 (27.5%)	46 (57.5%)	80 (100%)
F47	12 (15%)	51 (63.75%)	79 (98.75%)	30 (37.5%)	51 (63.75%)	67 (83.75%)	80 (100%)

Noted: black means the two factors are related, otherwise means unrelated.

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
