# Peer review of "Critical Factors of Promoting Design for Safety in China’s Subway Engineering Industry"

_ijerph, 2020, doi:10.3390/ijerph17103373_

Round 1

Reviewer 1 Report

The paper aims to identify key factors of promoting Design for Safety in China.

To my opinion the paper is interesting, but it needs major improvements to be accepted. In the present form it cannot be accepted.

Besides contents improvements that I will suggest below, the paper needs a strong and extensive editing of English language and style. Sentences do not sound, typos are present and formal language errors as well (i.e. "it's", "what's" etc..), Table 6 is not properly explained (what is red/black?), Figure 6 does not have the scale included/explained in caption. Before acceptance, those aspects need to be fixed.

Content issues:

  1. The focus of the research is not clear: the title suggests a focus on all industrial domains. Later in the paper, mainly construction (or underground constructions??) are considered. The authors need to choose a focus. To my opinion only the underground should be the one due to the majority of surveys.
  2. The abstract: the current one is not acceptable. An abstract should provide tell what the paper is all about. At the moment is an introduction to the problem, that should be placed in the Introduction. Please rewrite in a more academic form.
  3. Research Question should also be placed in some ways in the abstract and not only the "research methodology" section.
  4. Literature Review should have more specific focus: which is the main research domain you want to explore? DfS / PtD in underground? Please include at least those references:
    • Labagnara D., Martinetti A., Patrucco M. (2013). Tunneling operations, occupational S&H and environmental protection: a Prevention through Design approach, American Journal of Applied Sciences, 11,1371-1377, 10.3844/ajassp.2013.1371.1377.

    • Borchellini, R., Cardu, M., Colella F., Labagnara D., Martinetti A., Patrucco M., Sandrin D., Verda V. (2013). A Prevention through Design Approach for the Environmental S&H Conditions and the Ventilation System at an Italian Underground Quarry. In Chemical Engineering Transactions, AIDIC - The Italian Association of Chemical Engineering, 32, 181-186, ISBN 9788895608235, ISSN: 1974-9791, doi: 10.3303/CET1332031.

  5. Questionnaire Survey: 4 survey responses were collected: 20% came from Academics or Research Institutions. How could they help in understanding the perception on DfS in industries? They should be taken out. It should be kind of obvious that researchers on DfS understand the value of DfS. Moreover, check the numbers of Facilitating factors. You stated that 28 FF were retained and you omitted 3. Then, 25 should be available. Table 3 has only 24. Why that?
  6. Reliability Analysis: in my opinion is missing the main goal of that. Consequently, the connection with the method you have chosen. It has to be better explained. 
  7. Research Analysis: 6 aspects have been mentioned but only 5 FF are listed in figure 1. It is mandatory to fix it.

Author ResponseBesides contents improvements that I will suggest below, the paper needs a strong and extensive editing of English language and style. Sentences do not sound, typos are present and formal language errors as well (i.e. "it's", "what's" etc..), Table 6 is not properly explained (what is red/black?), Figure 6 does not have the scale included/explained in caption. Before acceptance, those aspects need to be fixed.
Answer:
Thanks for your suggestions. The red color in Table 6 indicates that less than 60% of the respondents believe there is an impact relationship between the two factors. While the black color indicates that more than 60% of respondents believe the facilitating factor can influence the impediment. I did not put a note about this in the paper before, now it has been added in the text and table. There are only 2 figures in the paper, and I have added explain about the reference line in figure 2 in the text.
Content issues:
1. The focus of the research is not clear: the title suggests a focus on all industrial domains. Later in the paper, mainly construction (or underground constructions??) are considered. The authors need to choose a focus. To my opinion only the underground should be the one due to the majority of surveys.
Answer:
Yes, the research is based on the National Natural Science Foundation of China (Project Number:51578144), it’s a project based on a subway project, mainly focused on subway related design, construction, and operation. Therefore, when we choose experts and questionnaire respondents, all the participants are from the subway engineering related to the firm’s staff or researchers. It is reflected in the text, but not clearly in the title of the article. To this end, the title is modified to Key Factors of Promoting Design for Safety in China's Subway Engineering Industry, to be more appropriate.
2. The abstract: the current one is not acceptable. An abstract should provide tell what the paper is all about. At the moment is an introduction to the problem, that should be placed in the Introduction. Please rewrite in a more academic form.
Answer:
According to your suggestion, the paper streamlines some of the current statuses in the abstract.
3. Research Question should also be placed in some ways in the abstract and not only the "research methodology" section.
Answer:
According to your suggestion, the paper has modified the abstract, add content about the research question and methodology.
4. Literature Review should have more specific focus: which is the main research domain you want to explore? DfS / PtD in underground? Please include at least those references:
• Labagnara D., Martinetti A., Patrucco M. (2013). Tunneling operations, occupational S&H and environmental protection: a Prevention through Design approach, American Journal of Applied Sciences, 11,1371-1377, 10.3844/ajassp.2013.1371.1377.
• Borchellini, R., Cardu, M., Colella F., Labagnara D., Martinetti A., Patrucco M., Sandrin D., Verda V. (2013). A Prevention through Design Approach for the Environmental S&H Conditions and the Ventilation System at an Italian Underground Quarry. In Chemical Engineering Transactions, AIDIC - The Italian Association of Chemical Engineering, 32, 181-186, ISBN 9788895608235, ISSN: 1974-9791, doi: 10.3303/CET1332031.
Answer:
The two references you have recommended, both are related to DfS in the underground, and have many similarities with subway projects I studies, so according to your suggestions, I added the two references in the literature review section.
5. Questionnaire Survey: 4 survey responses were collected: 20% came from Academics or Research Institutions. How could they help in understanding the perception on DfS in industries? They should be taken out. It should be kind of obvious that researchers on DfS understand the value of DfS. Moreover, check the numbers of Facilitating factors. You stated that 28 FF were retained and you omitted 3. Then, 25 should be available. Table 3 has only 24. Why that?
Answer:
In this study, 31 facilitating factors were obtained through literature reading. After the expert interview, 3 factors (F29, F36, and F48) were omitted, and 28 factors retained. Table 3 shows the results of the interview, and there are 31 factors, 3 factors in red are omitted. According to the results of the questionnaire survey, F18 and F28 were omitted, and 26 factors retained finally. The results were showing in table 8 (the two omitted factors were marked in red). According to your suggestion, some of the results from the academic or research institutions were added in the analysis section.
6. Reliability Analysis: in my opinion is missing the main goal of that. Consequently, the connection with the method you have chosen. It has to be better explained.
Answer:
I think the reliability analysis is an analysis of the consistency of the questionnaire results, and SPSS 22.0 can directly perform the analysis. According to the SPSS22.0, I should consider three factors, namely α coefficient, CITC, and item deleted α coefficient. Therefore, this study directly conducted a reliability analysis of the results of the questionnaire using SPSS 22.0. Theα coefficient values obtained were 0.912. all the item’s CITC were higher than 0.3, and all the item deleted α coefficient is not significantly higher than the α coefficient, so this study considers the reliability of the questionnaire to be acceptable. But the analysis of CITC and "term deleted alpha coefficient" is not reflected in the paper before, so based on your suggestions, I have added relevant content in the reliability analysis section.
7. Research Analysis: 6 aspects have been mentioned but only 5 FF are listed in figure 1. It is mandatory to fix it.
Answer:
I’m sorry for the wrote mistake, there are only five aspects in the analysis, which are four facilitating factors and one impediment factor. It has been corrected now.

Reviewer 2 Report

Dear authors.

I appreciate the opportunity to review this manuscript.This manuscript aims to identify the impediments and facilitating factors, and the relationships between the two factors, in order to help CCI introducing DfS successfully.The background of this manuscript was followings.
・It is of importance that safety should be prioritized to prevent accidents with a large number of casualties and economic losses during the construction and operation stages.
・It is significance that 60% of safety accidents occurred in engineering were related to design.
・China has not successfully implemented DfS, and the accident incident rate has not also been a definite decrease.
・Introducing DfS in China, especially in CCI, is an important step in the implementation of DfS during the world’s construction industry, and the most important thing is how to rapidly introduce DfS on the basis of those succeed countries.
・Previous literature pointed out that understanding the impacts, obstacles, and facilitators of DfS implementation is helpful for institutions or organizations in accepting and spreading DfS.
・It is the significance of analyzing the impediment and facilitating factors in CCI, and the relationship between them, in order to facilitate CCI concentrate its efforts on the main issues in the process of promoting DfS and avoid detours based on the taking subway construction in China.

The above contents are able to contribute to reducing the victims, the casualties and economic losses from the accidents.
I think researchers, administrators and planners related to the safety management of local municipalities would find this paper of interest if this manuscript has the possibility of new findings based on the analysis.However, this manuscript still has many critical or mainor problems as indicated below.

Critical problems

The conclusions of this manuscript were extremely shallow. This manuscript is not well discussed, and therefore, the conclusions section is not meaningful.

Specifically, the authors concluded that "The research revealed that designers’ lack of DfS related knowledge, together with a lack of guidelines and tools are the biggest obstacles for DfS implementation."

However, it is extremely shallow because this points of view are already pointed out by many previous studies .

Therefore, the authors should conclude about "What are the effective guidelines and tools for the DfS implementation in China?"
The international readers would like to know this point of view.

In addition, I would like the authors to indentified the novelty as scientific journals based on the comparing the previous literatures.The lack of above the point in the conclusion section is critical problems as publishing a scientific journal.

Minor problems

>3.2. Factor System Screening
If the authors use semi-structured interview, it is necessary to explain about ethical aspect for interviwees.>3.2.2.1. Content of Questionnaire Survey
>3.2.2.2. The selection of investigators
>3.2.2.3. The final factor system screening
These sections were different font size and styles, please confirm the guidelines.
If the authors find the problem based on the guidelines, please correct the style of these sentences.

>4. Research Results
I would strongly recommend that the authors explain about the basic information of resupondant in figures.

>4.3.3. The relationship between the facilitating factors and impediments
I would strongly recommend that the authors analyze the results based on classified by basic information. For example, I would like to know the "Are there any differences of the relationships of by classification of the "type of work conducted within the respondent’s firm" and "the personal work experience"."

I hope that my comment is very useful for the improvement of the manuscript.

Author Response

Critical problems The conclusions of this manuscript were extremely shallow. This manuscript is not well discussed, and therefore, the conclusions section is not meaningful.

Specifically, the authors concluded that "The research revealed that designers’ lack of DfS related knowledge, together with a lack of guidelines and tools are the biggest obstacles for DfS implementation."

However, it is extremely shallow because this points of view are already pointed out by many previous studies.

Therefore, the authors should conclude about "What are the effective guidelines and tools for the DfS implementation in China?"
The international readers would like to know this point of view.

In addition, I would like the authors to indentified the novelty as scientific journals based on the comparing the previous literatures.

Answer:

About your suggestions, before this research, we made a comparison between China and countries that have successfully implemented DfS in the status of tools, guidelines, policy, education, and practice, but I have submitted it in a Chinese journal (in the process of reviewing). To avoid repetition, the relevant content is lacking in this paper. In response to your suggestions, I have added some relevant content in the research analysis section. About the conclusion section, I just summarized some of the results of the questionnaire, and more detailed content is in the analysis part. Although there have been international studies that have concluded similar conclusions, most of the previous studies have targeted the population of countries that have implemented DfS, instead of people in China, which have not implemented DfS and most people have been exposed to DfS for the first time, so I think this research is still meaningful.

Minor problems

>3.2. Factor System Screening
If the authors use semi-structured interview, it is necessary to explain about ethical aspect for interviwees.

Answer:

Your suggestion was taken into consideration when we conducted the interview. When the expert was selected, firstly, we gave a brief introduction to them about our research, and then consulted the interviewed experts if they would like to participate and have time to participate. We will not force experts to participate if they do not want to.

>3.2.2.1. Content of Questionnaire Survey
>3.2.2.2. The selection of investigators
>3.2.2.3. The final factor system screening
These sections were different font size and styles, please confirm the guidelines.
If the authors find the problem based on the guidelines, please correct the style of these sentences.

Answer:

The font size and styles in these 3 sections have some mistake, I have modified it, thanks for your suggestions.

>4. Research Results
I would strongly recommend that the authors explain about the basic information of resupondant in figures.

Answer:

4 basic information was asked in the questionnaire, for the options are more classified, I only integrated some of them in the text before, now, I have been listed them in figures according to your suggestions.

>4.3.3. The relationship between the facilitating factors and impediments
I would strongly recommend that the authors analyze the results based on classified by basic information. For example, I would like to know the "Are there any differences of the relationships of by classification of the "type of work conducted within the respondent’s firm" and "the personal work experience"."

Answer:

The research has previously analyzed the results based on the basic information, however, for the sample size of this study is small and there are less significant conclusions, so it was not shown in this paper before, now I increased some analysis by the type of work conducted within the respondent’s firm in the analysis section.

Round 2

Reviewer 1 Report

Dear authors,

thanks for your work. My suggestion has been addressed. In my opinion now the paper is almost ready for pubblication.

Just 2 minor points.

-The following ref. has been added. But in the reference list is not written correctly. Please you this notation:

• Borchellini, R., Cardu, M., Colella F., Labagnara D., Martinetti A., Patrucco M., Sandrin D., Verda V. (2013). A Prevention through Design Approach for the Environmental S&H Conditions and the Ventilation System at an Italian Underground Quarry. In Chemical Engineering Transactions, AIDIC - The Italian Association of Chemical Engineering, 32, 181-186, ISBN 9788895608235, ISSN: 1974-9791, doi: 10.3303/CET1332031.

-Please check the abstract: the first sentence has a typos:

"For DfS (Design for Safety) can greatly reducing the engineering safety risk, decreasing the economic and social losses caused by the safety accident".

Author Response

thanks for your work. My answer has been addressed.

Question 1:

-The following ref. has been added. But in the reference list is not written correctly. Please you this notation:

  • Borchellini, R., Cardu, M., Colella F., Labagnara D., Martinetti A., Patrucco M., Sandrin D., Verda V. (2013). A Prevention through Design Approach for the Environmental S&H Conditions and the Ventilation System at an Italian Underground Quarry. In Chemical Engineering Transactions, AIDIC - The Italian Association of Chemical Engineering, 32, 181-186, ISBN 9788895608235, ISSN: 1974-9791, doi: 10.3303/CET1332031.

Answer

Thanks for your suggestions. The reference was directly generated by the NoteExpress in this paper. However, this reference was wrongly written when at that time, so it caused the subsequent reference writing mistake. Now it has been corrected.

Question 2:

-Please check the abstract: the first sentence has a typos:

"For DfS (Design for Safety) can greatly reducing the engineering safety risk, decreasing the economic and social losses caused by the safety accident".

Answer

Thanks for your kindly reminder, it has been corrected in the paper.

Reviewer 2 Report

Thank you for your response.

However, the authors' response may contain ethical issues and does not accept as a scientific paper.

The findings of the manuscript are specific to China and there is no positioning as an international perspective.
In addition, since the authors have already submitted important findings related to the manuscript to Chinese domestic journals, it is not necessary to publish them in this journal.
It is considered unethical to split the results of a study that can be published as a single research paper.

Author Response

The authors' response may contain ethical issues and does not accept as a scientific paper.

The findings of the manuscript are specific to China and there is no positioning as an international perspective.
In addition, since the authors have already submitted important findings related to the manuscript to Chinese domestic journals, it is not necessary to publish them in this journal.
It is considered unethical to split the results of a study that can be published as a single research paper.

Answer

Thanks for your suggestions, but I think you have some mistakes to understand the paper.

Firstly, this paper is mainly based on the National Natural Science Foundation of China, “DFS-based Intelligent Pre-control Methodologies Aiming at Reducing Safety Risks during Lifecycle of Subway Engineering” (Project No. 51578144). There is no doubt one research can have more than one outcome, so I think submitted one of the findings in this paper has no affection for submitted other findings in another journal. The two papers just have a different theme, there is no separation and overlap between the two papers, and they are different, therefore, there is no unethical for me and this paper. In contrast, I think that one paper must have one theme, if I overlap the two papers into one, it is not correct, and the paper will have more than one theme. Besides, the different journal has different readers, I submit the other paper in a Chinese journal for the other is about literature review about DfS (different with content in this paper), and I want to introduce and promote DfS concept to more Chinese researchers and practitioners. If you have any questions about the other paper, I can submit the other paper’s manuscript to you for the check.

Besides, I also think you have some mistakes to understand China’s construction industry market in the mainland and overseas. Even though there have some countries successfully implement DfS, most of the countries in the world had not realized the usefulness of DfS. According to the ENR, like what I have been written in the paper, Chinese company have a large amount of the market in the world and have rapid growth rate, the acceptance of DfS in China will have no doubt influence the implementation in the whole world, because DfS will be used by oversea projects constructed by Chinese companies, and projects in China but constructed by other countries. Therefore, I think the paper has enough international perspective.
